

# Remineralisation changes dominate oxygen variability in the North Atlantic

Rachael N.C. Sanders[1,2], Elaine L. McDonagh[1,3], Siv K. Lauvset[1], Charles E. Turner[4], Thomas W.N. Haine[5], Nadine Goris[1], and Richard Sanders[1]

[1]NORCE Research AS, Bjerknes Centre for Climate Research, Bergen, Norway
[2]Now at British Antarctic Survey, Cambridge, UK
[3]National Oceanography Centre, Southampton, UK
[4]ACCESS-NRI, Australian National University, Canberra, Australia
[5]Department of Earth & Planetary Sciences, Johns Hopkins University, Baltimore, Maryland, USA

**Correspondence:** Rachael Sanders (racnde@bas.ac.uk)

**Abstract.** Oxygen is fundamental to ocean biogeochemical processes, with deoxygenation potentially reducing biodiversity, and disrupting biogeochemical cycles. In recent decades, the global ocean oxygen concentration has been decreasing, but this decrease is underestimated in numerical ocean models by as much as 50%. Mechanisms responsible for this deoxygenation include solubility-driven deoxygenation driven by ocean warming, and changes in the amount, rates and spatial patterns of

remineralisation. However, the magnitude of change in oxygen due to each process is currently unclear. Here, we use a new method to decompose oxygen change into its constituent parts by linking each process to concomitant changes in temperature and dissolved inorganic carbon. Using observations across a repeated section of the North Atlantic at 24.5°N, we show that the consistent oxygen decrease observed since 1992 in the upper 2000 m has been dominated by an increase in remineralisation-related oxygen-consumption. While warming-driven solubility changes have a much smaller impact on the upper ocean in

comparison, the impact has trebled in the past twenty years, suggesting they will become an increasingly significant driver of deoxygenation with future warming. Remineralisation-related oxygen consumption peaks at a depth of approximately 600 m, where it is responsible for up to 70% of the total deoxygenation. This remineralisation-driven change may be caused by a change in the supply of biological material to depth, a change in circulation leading to change in the residence time of water in the North Atlantic and hence the accumulation of the remineralised oxygen deficit, or a combination of both. While this study

does not determine the exact cause, previously little change in productivity in has been observed in the region, suggesting ocean circulation is indirectly driving the majority of deoxygenation in the Subtropical North Atlantic, via a non-local change in remineralisation.

## 1 Introduction

Oxygen is a fundamental component of biogeochemical processes in the ocean. A change in the ocean oxygen content can

result in major changes in biodiversity, and biogeochemical cycles (Gruber, 2011; Morée et al., 2023). Since the mid-20th century, both open ocean and coastal waters have experienced a marked decline in oxygen concentration (Stramma et al., 2008;





Diaz and Rosenberg, 2008; Keeling et al., 2010; Helm et al., 2011), with approximately a 2% reduction in open ocean oxygen concentrations over the past 50 years (Schmidtko et al., 2017). However, there is disagreement between observations of oxygen change and results of Coupled Model Intercomparison Project (CMIP) model simulations (Oschlies et al., 2017; Takano et al.,

2023; Abe and Minobe, 2023), both in terms of magnitude and spatial patterns, with models simulating a decline in the global ocean oxygen inventory that is only approximately half of the observed deoxygenation (Oschlies et al., 2018). While there are many different processes driving deoxygenation, some of which are not necessarily accurately represented by the models, it is unclear which of these processes is responsible for the too conservative decline in the models.

Generally, oxygen in the surface ocean is always close to saturation due to rapid exchange with the atmosphere. While some

of this oxygen enters the interior ocean, deoxygenation occurs with depth due to an imbalance between ventilation and the consumption of organic matter at depth (Levin, 2018). Oxygen concentrations are sensitive to ocean warming via temperature-driven changes in solubility. Ocean oxygen and heat content are therefore highly correlated, with a sharp rise in heat content and concurrent deoxygenation observed from the mid 1980s (Ito et al., 2017). While warming-induced solubility changes may be a significant driver of deoxygenation in the upper ocean (Shaffer et al., 2009; Helm et al., 2011), the total ocean oxygen

loss has been shown to be around 3-4 times higher than expected from the direct impact of warming alone (Oschlies, 2021), indicating that other mechanisms play a significant role. It has been hypothesised that the disparity between observational and modelled oxygen decline is due to uncertainty in either historic atmospheric forcing data, or the representation of biogeochemical processes in models (Oschlies et al., 2017). Projections show an acceleration of the observed deoxygenation with ongoing climate change (Bopp et al., 2013; Keeling et al., 2010), so understanding the causes of disparities between model results and

observations is imperative for accurate predictions of future global oxygen changes.

Warming ocean temperatures also impact oxygen concentrations indirectly via altering key terms such as stratification or biological processes (Breitburg et al., 2018; Stendardo and Gruber, 2012; Keeling et al., 2010). Acceleration of oxygen consumption rates leads to changes in the depth distribution of oxygen loss as the respiration of organic matter occurs at shallower depths (Brewer and Peltzer, 2017; Oschlies, 2021). Increases in ocean stratification and freshwater fluxes account for a signifi-

cant amount of the remaining global oxygen loss, via changes in ventilation and nutrient supply (Bopp et al., 2013; Long et al., 2016). Circulation changes also influence oxygen concentration, driving spatial variation in mixed layer depths, and altering water mass residence times (Palter and Trossman, 2018). In deep waters, changes in oxygen are predominantly driven by a change in the balance between ventilation, and the consumption of oxygen via the respiration of sinking particulate organic matter (Matear and Hirst, 2003).

To allow for a better understanding of these disparities, we use a new method to apportion any oxygen changes due to each potential driving mechanism, by linking those changes to related changes in the driving mechanisms of temperature and the concentration of dissolved inorganic carbon (DIC). This is based on a method previously used to divide temperature changes into excess changes and redistribution (Turner et al., 2022). Here, excess changes are defined as oxygen being added/removed from the system via a change in solubility, and redistribution is of the background oxygen already in the system. We also

consider two additional drivers: changes due to remineralisation, and a change in the disequilibrium between the surface ocean and atmosphere when the oxygen was first absorbed, leading to changes in air-sea fluxes. We apply this method to observations





from the subtropical North Atlantic, at 24.5°N, the most frequently surveyed transoceanic section, with six cruises over 23 years between 1992 and 2015. Over this period, the region has experienced a notable oxygen decline (Stendardo and Gruber, 2012), increased temperature and salinity (Turner, 2024), increased accumulation of anthropogenic carbon (Guallart et al., 2015), and

trends in cross-section transport, with a decrease in the northward flow of Antarctic Intermediate Water (Hernández-Guerra et al., 2014).

## 2  Methods

### 2.1  Data and Processing

We use GLODAPv2.2002 (Lauvset et al., 2022; Olsen et al., 2016; Key et al., 2015) potential temperature, DIC, and oxygen

bottle measurements, interpolated onto the GO-SHIP Easy Ocean (Katsumata et al., 2022) A05 section. The A05 hydrographic section is located at approximately 24.5°N, spanning 80-13°W across the subtropical North Atlantic (Fig. 1a). Using DIVA (Data-Interpolating Variational Analysis) gridding techniques (Barth et al., 2014), we interpolate the bottle data onto a regular grid with 651 vertical levels and 670 longitude points, using the same methods as Turner (2024). The section was sampled in six cruises, conducted between 1992 and 2015 (Table A1). We quantify changes in oxygen concentration, potential temperature,

and DIC by computing the difference between measurements obtained during the initial cruise in 1992 and those obtained in each subsequent cruise (e.g., 1992-1998, 1992-2004, and so forth).

### 2.2  Processes Driving Oxygen Change

We define the total change in oxygen as the sum of four distinct changes: 1. a change in the excess oxygen, 2. redistribution of the background field, 3. change in the amount of remineralisation, and 4. change due to variability in sea surface disequilibrium.

Excess changes are those due to changes in sea surface temperature which alter solubility. The redistribution of the oxygen already in the water column is a result of circulation changes, or reorganization of the local oxygen structure, and so will sum to zero globally. It is a result of the local movement or change in the distribution of water masses, and occurs on all time scales associated with local variability in circulation, which is well measured at the A05 section (Frajka-Williams et al., 2019). Our assumption, following Turner (2024), is that these redistributed changes are linked by local stratification in theta-oxygen-DIC

space. While we determine the interannual changes in redistribution well, we do not resolve them well on shorter timescales, as they vary on timescales much faster than the nominal GO-SHIP repeat of ~5 years at this latitude. However, the water mass properties we are interested in change more slowly, and so are resolved by the ~5 year frequency.

Changes in remineralisation can be either due to a change in the rate export of organic matter leading to a higher rate of remineralisation at depth, or due to changes in the spatial patterns of remineralisation, i.e. because the same amount of

remineralisation is occurring but at a different point in space. The final, disequilibrium change is due to the change in saturation levels when the water mass was initially subducted, which changes the potential amount of oxygen that can be absorbed by the surface ocean.



**Figure 1.** a) Map illustrating the A05 cruise transect overlaid on background bathymetry (m) (NOAA National Centers for Environmental Information, 2022); the circle denotes 45°W, the midpoint of the transect at the approximate position of the mid-Atlantic ridge. b-d) The average potential temperature (°C), DIC ($\mu$mol kg$^{-1}$) and oxygen ($\mu$mol kg$^{-1}$) profiles east of 45°W, for the six cruises between 1992 and 2015, and e-g) the profiles averaged over the western half of the section.



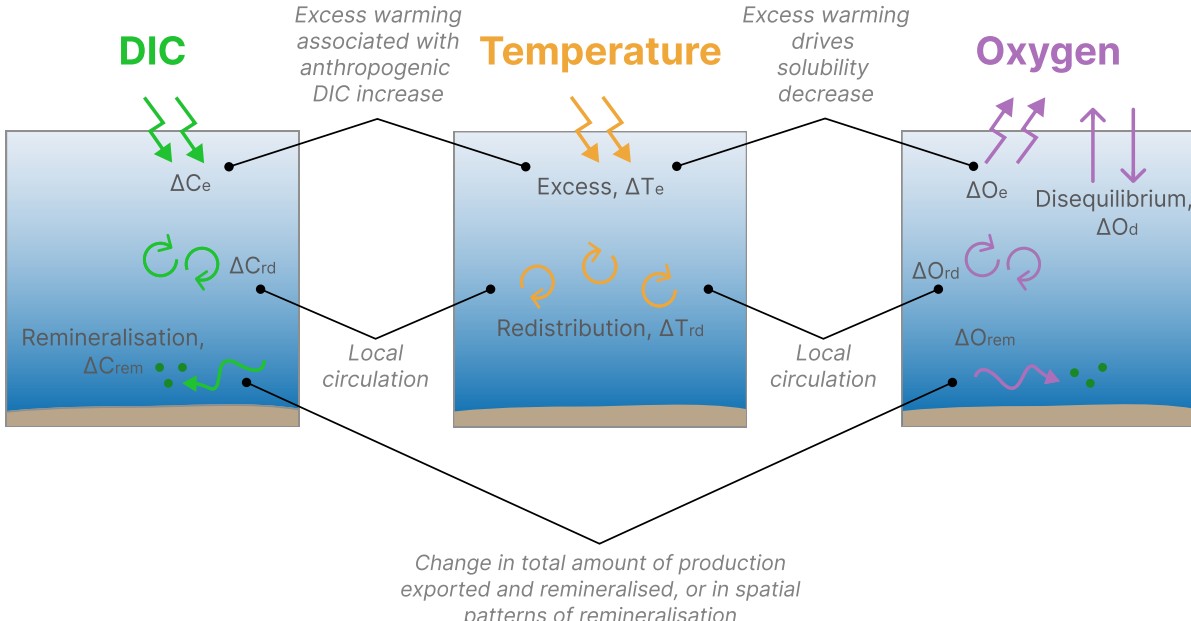

**Figure 2.** Schematic illustrating the links between the different processes driving changes in oxygen, temperature, and DIC, along with the coefficients used to link the processes.

These mechanisms are shown in a schematic in Figure 2. By linking each of these mechanisms to simultaneous changes in temperature and DIC, we develop a set of equations that can be solved to calculate the magnitude of each process. The total temperature change is the sum of only excess and redistributed changes, while total DIC change is defined as a sum of the excess, redistributed and remineralisation changes. Excess DIC change is dominated by the anthropogenic increase (Turner, 2024), and is associated with an associated temperature change. Isolating this excess change means that we can also calculate the remaining redistribution component of temperature change. Since the redistribution of all three properties is driven by local changes in water masses, we can use the redistribution of temperature to calculate the redistribution of DIC and oxygen. The remaining DIC change, due to the production of DIC during remineralisation, is connected to the associated loss of oxygen. The disequilibrium oxygen term remains weakly constrained as it is not directly linked to another process. A disequilibrium term is not included for DIC because, while disequilibrium does affect surface $CO_2$ concentrations, this term is instead incorporated in the coefficient relating any excess DIC and temperature changes, described in Section 2.4.

The total change in oxygen ($\Delta O$), potential temperature ($\Delta T$), and DIC ($\Delta C$) is thus described by the following equations:

$$\Delta O = \Delta O_e + \Delta O_{rd} + \Delta O_{rem} + O_d, \tag{1}$$





$$\Delta T = \Delta T_e + \Delta T_{rd}, \tag{2}$$

$$\Delta C = \Delta C_e + \Delta C_{rd} + \Delta C_{rem}, \tag{3}$$

where the subscript $e$, $rd$, $rem$, and $d$ relate to excess, redistribution, remineralisation, and disequilibrium change, respectively. Each of these contributions are currently unknown. By defining coefficients that link the individual components of temperature,
DIC, and oxygen change, we can form a system of equations that can be solved to obtain the magnitude of each process driving oxygen change. We only look at the upper 2000 m of the water column because the changes in the deeper ocean over the period that the A05 section has been occupied are smaller than the accuracy of the GLODAP data, particularly for DIC.

## 2.3   System of Equations

We can formulate a system of equations in the matrix form $\mathbf{Ax} = \mathbf{b}$, where $\mathbf{A}$ is a matrix of coefficients linking the oxygen
change terms to those in DIC and potential temperature, and $\mathbf{b}$ is a vector containing the total change in each property since 1992. This under-determined system of equations can then be solved, using a weighted least squares fit method, to obtain $\mathbf{x}$, i.e. the excess, redistributed, remineralisation, and disequilibrium change in oxygen at each point:

$$\begin{bmatrix} \eta & \beta_o & 0 & 0 \\ \frac{\eta}{\alpha} & \frac{\beta_o}{\beta_c} & \gamma & 0 \\ 1 & 1 & 1 & 1 \end{bmatrix} \begin{bmatrix} \Delta O_e \\ \Delta O_{rd} \\ \Delta O_{rem} \\ \Delta O_d \end{bmatrix} = \begin{bmatrix} \Delta T \\ \Delta C \\ \Delta O \end{bmatrix}. \tag{4}$$

The full derivation of these equations is shown in the appendix, and the coefficients in $\mathbf{A}$ are calculated as follows.

## 2.4   Links between Excess Changes

Excess changes in oxygen and DIC are both linked to excess changes in temperature, via different mechanisms. Changes in surface heat fluxes lead to changes in solubility, and therefore oxygen concentration, while an almost linear relationship exists between ocean heat and carbon content, both on local and global scales (Bronselaer and Zanna, 2020), due to the oceanic uptake of heat and carbon as a response to anthropogenic $CO_2$ emissions.
Excess oxygen changes are due to temperature-driven change in solubility, a relationship that is close to linear over the observed temperature ranges in the A05 section. This relationship is described by the coefficient $\eta$, the gradient of the temperature-oxygen solubility curve, assuming 100% saturation and using the temperature during the 1992 cruise as the initial temperature of the curve. Within the A05 section, the coefficient has a range of -0.33 to -0.12°C ($\mu$mol/kg)$^{-1}$, and increases with depth (Fig. 3b). The effect of oxygen under-/over-saturation is not incorporated in this coefficient since the saturation level is not





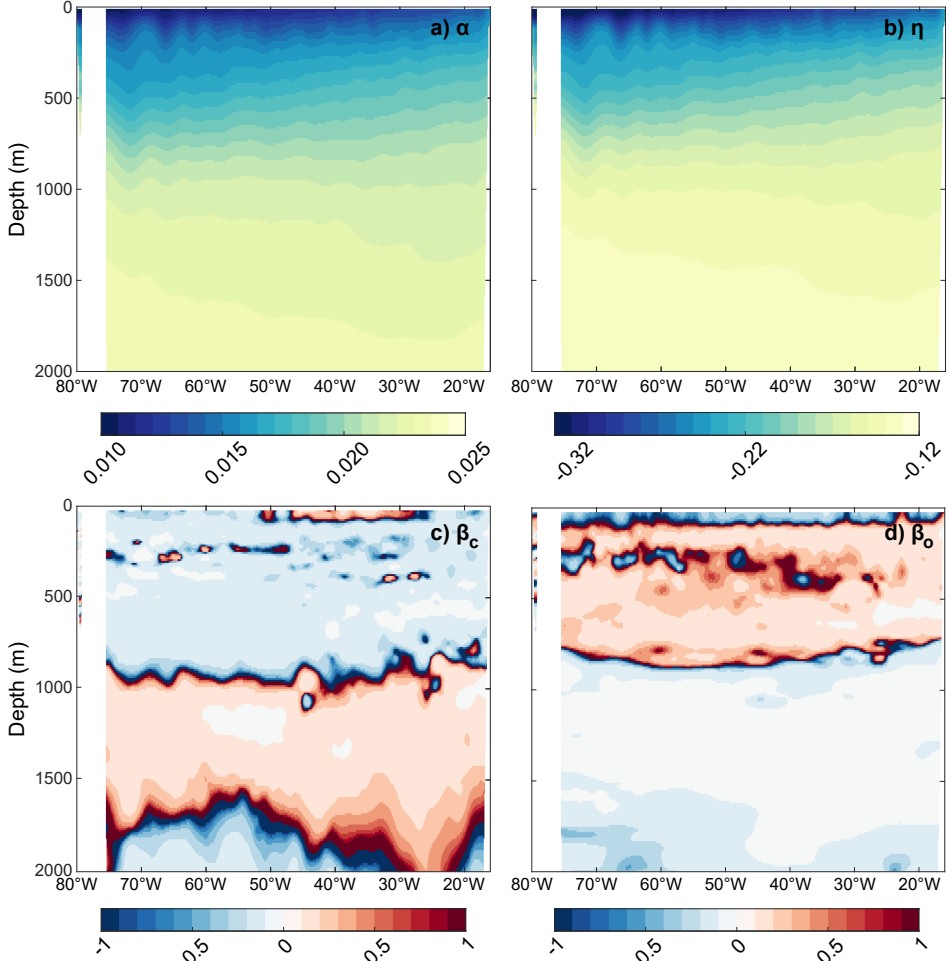

**Figure 3.** a) The coefficient $\alpha$ (°C $(\mu\text{mol/kg})^{-1}$), representing the relationship between excess potential temperature changes and excess DIC change. b) The solubility coefficient $\eta$ (°C $(\mu\text{mol/kg})^{-1}$), linking excess temperature change and excess oxygen change. c,d) The coefficients $\beta_C$ and $\beta_O$ (°C $(\mu\text{mol/kg})^{-1}$), linking redistribution-driven changes in temperature with redistribution-driven changes in DIC and oxygen, respectively. Each coefficient is calculated from the 1992 occupation of the A05 section, as this is the earliest occupation and so least impacted by excess changes.

known. Instead, adjustments for saturation effects are addressed through the inclusion of disequilibrium change as a distinct oxygen change term. The relationship between excess oxygen and temperature change is described by Equation 5:

$$\Delta T_e = \eta \Delta O_e. \tag{5}$$

To describe the relationship between excess temperature and excess DIC, we use a temperature-dependent coefficient, $\alpha$, (Turner et al., 2022). Equation 7 defines the relationship between the excess temperature and excess DIC terms. This coefficient was approximated as uniform via Equation 6, following Turner (2024). The coefficient $\alpha$ varies between 0.010 and 0.024°C



$(\mu mol/kg)^{-1}$ within the A05 section in 1992, and increases with depth (Fig. 3a), similar to $\eta$. We use the initial (1992) values for $\alpha$, however temporal variability in the coefficient is minimal over these timescales, with a mean difference on the order of $10^{-6}$ $\mu mol/kg$. Since $\alpha$ describes the observed relationship between the excess component of DIC and temperature, it also takes into account any impact that the change in $CO_2$ saturation has on the surface flux. Separating this effect would be non-
trivial and add extra unknowns into the equation, so it is simply included as part of the excess term and no DIC disequilibrium term is introduced:

$$\alpha = 0.025 - 0.0005T, \tag{6}$$

$$\Delta T_e = \alpha \Delta C_e. \tag{7}$$

## 2.5 Link between Redistributed Terms

The redistribution coefficients $\beta_C$ and $\beta_O$ relate redistribution driven changes in temperature to redistribution driven changes in DIC and oxygen, at a given geographical location. Each coefficient is approximated by the ratio of stratification between temperature and DIC/oxygen, following Turner (2024):

$$\beta_C = \frac{\partial T}{\partial z} \left( \frac{\partial C}{\partial z} \right)^{-1}, \tag{8}$$

$$\beta_O = \frac{\partial T}{\partial z} \left( \frac{\partial O}{\partial z} \right)^{-1}. \tag{9}$$

Within the A05 section, $\beta_C$ has a mean value of -0.043°C $(\mu mol/kg)^{-1}$ (Fig. 3c), and is primarily negative in the upper 1000 m as temperature decreases while DIC increases with depth. The coefficient then transitions to positive at ∼1000 m as DIC concentration reaches a maximum and begins to decrease slightly. $\beta_O$ has a mean of -0.048°C $(\mu mol/kg)^{-1}$ (Fig. 3d), and is negative in the upper 100 m as temperature and oxygen are relatively constant. The coefficient then shifts to positive as both the oxygen concentration and temperature decreases, until a depth of around 800 m, when oxygen reaches a minimum
and begins to decrease again, driving a negative $\beta_O$. At this depth, as the coefficient changes sign, redistribution is effectively zero. The relationship between redistributed changes in temperature, DIC, and oxygen can then be expressed by the following equations:

$$\Delta T_{rd} = \beta_C \Delta C_{rd}, \qquad\qquad \Delta T_{rd} = \beta_O \Delta O_{rd}. \tag{10}$$

## 2.6 Link between Remineralisation Terms

We define the relationship between changes in remineralisation in oxygen and those in DIC using the Redfield ratio of 106C:-138O (Redfield, 1934, 1958), generally considered to be the approximate average ratio of change during remineralisation. The





|  | Initial guess | Weighting |
|---|---|---|
| Excess oxygen change, $\Delta O_e$ ($\mu$mol kg$^{-1}$) | -0.118 [a] | 0.111 [a] |
| Redistributed oxygen change, $\Delta O_{rd}$ ($\mu$mol kg$^{-1}$) | 0 [b] | 4.372 [a] |
| Remineralisation oxygen change, $\Delta O_{rem}$ ($\mu$mol kg$^{-1}$) | 0 [b] | 7.0 [a] |
| Disequilibrium oxygen change, $\Delta O_d$ ($\mu$mol kg$^{-1}$) | 0 [b] | 3.0 [a] |
| Total temperature change, $\Delta T$ (°C) | - | 0.005 [c] |
| Total DIC change, $\Delta C$ ($\mu$mol kg$^{-1}$) | - | 4.0 [c] |
| Total oxygen change, $\Delta O$ (%) | - | 0.4 [c] |

**Table 1.** The values used to regularise each variable in Equation 4 when calculating the decomposition of the oxygen change. [a]The weightings and initial guess for each individual oxygen components defined as the expected mean and standard deviation respectively, taken from literature. [b]When no previous information is available, the initial guess is set to zero. [c]The weightings for the total variable changes come from the approximate consistency between cruises in GLODAP, with the oxygen weighting changing spatially due to being given as a percentage.

change in remineralised oxygen is therefore described in Equation 11, where $\gamma = \frac{-106}{138} \approx -0.77$:

$$\Delta C_{rem} = \gamma \Delta O_{rem}. \tag{11}$$

The ratio $\gamma$ between oxygen and DIC is tightly constrained based on interior tracer fields, and while there will be some spatial
changes in this ratio, within the Atlantic, the remineralisation ratio between oxygen and carbon can be considered constant with depth (Li and Peng, 2002).

## 2.7   System regularisation

Once the coefficients are calculated for each individual gridpoint in the A05 section, they can be input into matrix $A$ and the system in Equation 4 can be solved via a weighted least squares fit approach. The magnitude of the oxygen change terms at
each point can then be added back into the equations relating oxygen change to temperature and DIC changes to compute the magnitude of each driver of temperature and DIC change at each point in the A05 section.

As Equation 4 is under-determined, an infinite number of solutions exist, and so we regularise the system with an initial guess of the solution and its estimated variance (Wunsch, 2006). These weightings are listed in Table 1. The **b** vector consisting of total changes in temperature, DIC and oxygen, is weighted by the inconsistency in each variable between GLODAP cruises
(Lauvset et al., 2022); this weighting determines the possible magnitude of the residual (i.e. the mismatch between the total oxygen change and the sum of the driving terms). Temperature and oxygen weightings are derived from the inconsistency between cruises within the Atlantic, while the global inconsistency in the GLODAP data product is used for DIC, as the Atlantic inconsistency is too high due to the influence of coastal ocean data.



The initial guess and weighting for the **x** vector, which consists of the changes in each individual oxygen process, are

assigned as the estimated mean, and variance of each component, respectively, when estimates of these values exist in the

literature. In instances where no previous information is available, we instead assign an initial guess of zero, but must still give

a non-zero value for the weighting in order to obtain a non-zero solution. In this case, when temporal variability data is not

available, we assume an estimate of the spatial variability from previous literature to be comparable. A weighting and initial

guess for excess oxygen change is approximated from excess temperature change along the A05 section (Zika et al., 2021) via

Equation 5. Weightings for remineralisation and disequilibrium changes are estimated from spatial variability available in the

literature (Cassar et al., 2021; Ito et al., 2004). We assume variability in the redistributed term to be similar to that of the total

oxygen change, and thus take the variance in the total change as the redistributed weighting.

## 3  Results

### 3.1  Total Oxygen Change in the A05 Section

The observations show an overall decline in total oxygen concentration in the upper 1000 m of the A05 section, which is

strongest at the surface (Fig. 4a-e), with the trend becoming more pronounced from 2010 onwards. The exception to this is in

the eastern side of the section, which shows an oxygen increase from around 200-1000 m eastward of 30 °W, likely related to

the influence of Mediterranean water and denitrification. Below 1000 m, there is no obvious trend, with areas of both positive

and negative oxygen change. At depths of 1000-2000 m an increase in oxygen between 40 and 65 °W is observed in most

years, except 1992-2010, when this region instead experiences a strong decrease.

To better understand the drivers of the average property changes across the A05 section, we focus on the central region

between 30–70°W. The average over this region emphasises the negative trend in total oxygen change in the upper 1000 m of

the water column (Fig. 5a), with deoxygenation reaching a maximum rate of -13.1 $\mu$mol kg$^{-1}$ between 1992 and 2010 at $\sim$600

m depth. As the average oxygen concentration decreases, the observations simultaneously show a strong DIC increase in the

upper ocean (Fig. 5g), reaching a maximum change of 32.4 $\mu$mol kg$^{-1}$ from 1992 to 2015 at $\sim$300 m. However, there is no

consistently increasing trend in the average temperature of the section over the same period (Fig. 5l).

### 3.2  Decomposition of Oxygen Change Across the A05 Section

There is large variability in the different driving mechanisms of oxygen, DIC and temperature change at different depths.

Averaging over different depths also shows high standard deviation within those depth ranges at depths greater than 1000 m

(Table 2). From 150-500 m depths, the total deoxygenation is dominated by change in remineralisation, the total DIC increase

is dominated by excess change, and slight warming is driven by excess change. The dominance of these processes becomes

less clear at depths of 500-1000 m. Remineralised oxygen change still makes up the majority of the deoxygenation, while for

DIC, the remineralisation term also becomes more significant, with an average of 3.42 $\mu$mol kg$^{-1}$ compared to a total DIC





| | | 150-500 m | 500-1000 m | 1000-2000 m |
|---|---|---|---|---|
| **Oxygen change ($\mu$mol kg$^{-1}$)** | Total | -7.82 $\pm$ 6.13 | -6.78 $\pm$ 5.46 | 0.44 $\pm$ 2.81 |
| | Excess | -1.49 $\pm$ 0.24 | -0.47 $\pm$ 0.45 | -0.075 $\pm$ 0.26 |
| | Redistributed | 0.01 $\pm$ 1.46 | 0.034 $\pm$ 1.42 | -0.13 $\pm$ 1.16 |
| | Remineralisation | -5.61 $\pm$ 3.92 | -4.46 $\pm$ 3.60 | 0.61 $\pm$ 1.43 |
| | Disequilibrium | -0.58 $\pm$ 1.88 | -1.55 $\pm$ 1.55 | 0.03 $\pm$ 0.72 |
| **DIC change ($\mu$mol kg$^{-1}$)** | Total | 28.19 $\pm$ 5.42 | 8.20 $\pm$ 7.55 | -0.37 $\pm$ 3.59 |
| | Excess | 20.53 $\pm$ 4.60 | 4.57 $\pm$ 4.66 | 0.51 $\pm$ 1.64 |
| | Redistributed | 0.19 $\pm$ 2.90 | -0.42 $\pm$ 1.71 | -0.01 $\pm$ 0.37 |
| | Remineralisation | 4.31 $\pm$ 3.01 | 3.42 $\pm$ 2.77 | -0.47 $\pm$ 1.10 |
| **Temperature change (°C)** | Total | 0.38 $\pm$ 0.33 | 0.18 $\pm$ 0.23 | 0.03 $\pm$ 0.10 |
| | Excess | 0.34 $\pm$ 0.06 | 0.09 $\pm$ 0.09 | 0.01 $\pm$ 0.04 |
| | Redistributed | 0.01 $\pm$ 0.31 | 0.07 $\pm$ 0.18 | 0.01 $\pm$ 0.05 |

**Table 2.** The total change in oxygen, DIC and temperature from 1992-2015, and the change due to each process, averaged over the central region of the A05 section and different depths, along with the standard deviation. The upper 150 m has not been included to avoid the influence of seasonal variability.





change of 8.20 $\mu$mol kg$^{-1}$. At this depth, the average temperature change is less than the standard deviation, with both excess

change and redistribution driving a similar size increase.

The patterns in the total oxygen change resemble those of the remineralisation change (Fig. 4p-t), with consumption of oxygen via remineralisation generally increasing over time in the upper 1000 m, indicating an increase in the total amount of remineralisation. The maximum oxygen remineralisation decrease occurs at a depth of ~600 m, reaching a maximum of -7.4 $\mu$mol kg$^{-1}$ by 2015 (Fig. 5d). The majority of remineralisation oxygen change is negative in the central part of the section,

with only a slightly positive average change below 1000 m, which is strongest between 1992 and 2004. The increasing total oxygen seen in the east of the section is also due to remineralisation changes. When averaged over the entire central region of the A05 section, remineralisation change is strongly correlated with total oxygen change (r$^2$ = 0.89).

Excess oxygen changes exhibit the clearest temporal trend, with increased deoxygenation since 1992, particularly in the upper 1000 m (Fig. 4f-j). Despite its smaller magnitude in comparison to remineralisation change, the term exhibits the largest

trend over time, owing to increasing excess temperature changes and their impact on oxygen solubility. Excess increase peaks at ~300 m, with the maximum deoxygenation at this depth trebling between 1992-1998 and 1992-2015, when it reaches -1.50 $\mu$mol kg$^{-1}$ (Fig. 5b).

Redistributed changes in oxygen are less consistent with time (Fig. 5c), reflecting regional circulation patterns with no apparent trends. The lines of near-zero redistribution close to the surface and at a depth of around 1000 m (Fig. 4k-o) are

simply due to the change in sign of the $\beta_o$ coefficient as it switches from positive to negative. At this depth, redistribution has no effect on oxygen because the depth gradient is zero.

Patterns of disequilibrium oxygen change are relatively similar to those of the total oxygen change, but with considerably lower absolute values (Fig. 4u-y). When averaged across the central region of the section, the maximum magnitude is only -2.37 $\mu$mol kg$^{-1}$ at depths of ~600 m between 1992-2010 (Fig. 5e). Since there is no direct link between the disequilibrium

change and any of the DIC or temperature terms, the term is weakly constrained, making analysis more challenging. The similarity between the total change is likely because it is a residual term, and so will have a larger magnitude when the total change is greater.

Fig. 6 shows the remineralisation and excess change over time, as the two terms with clear trends. Averaging the changes over the upper 1000 m emphasises the relationship between total oxygen change and the remineralisation component. In both

time series, maximum deoxygenation occurs in 2010, before decreasing slightly again. Solubility-driven deoxygenation has a much smaller but fairly constant decrease across the time series in the upper 1000 m. At depths of 1000-2000 m (Fig 6b), there is no obvious trend in total oxygen change across the section, however, the total change and remineralisation change are still highly correlated, with the same patterns of increasing and decreasing oxygen over time.

### 3.3 Temperature and DIC Change in the A05 Section

The temperature change in the central region of the A05 section is dominated by redistribution (Fig. 5n). While no clear trends are observed in the redistributed component, the excess component exhibits a steadily increasing trend, particularly pronounced



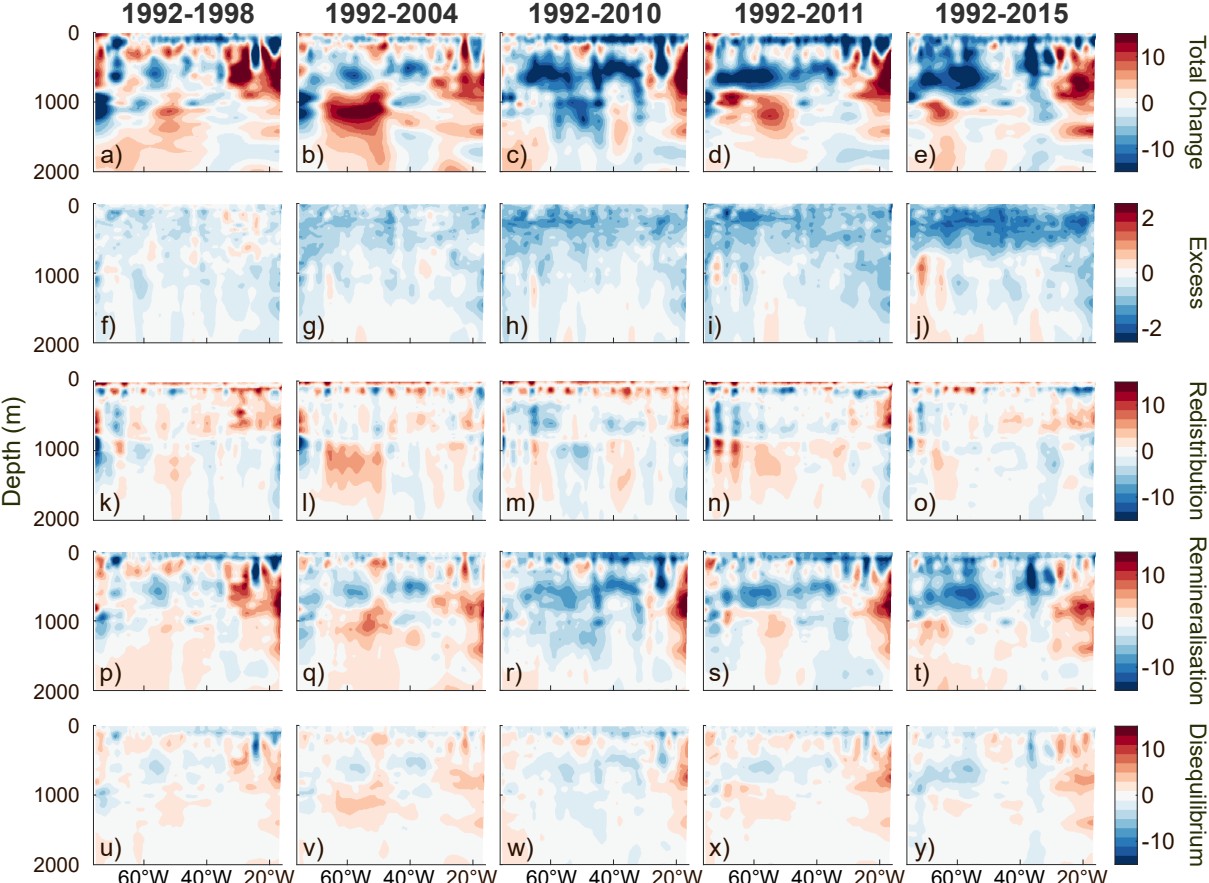

**Figure 4.** The oxygen decomposition results for the upper 2000 m of the A05 section: each plot shows the oxygen change ($\mu$mol kg$^{-1}$) between 1992 and each subsequent cruise year. a-e) The total oxygen change since 1992, f-j) the excess change, k-o) the redistributed change, p-t the remineralisation change), and u-y) the disequilibrium change. The surface 150 m is omitted from a) to rule out any effects due to seasonal variability. Note the varying colour scales.

in the upper 1000 meters (Fig. 5m). This leads to decreased solubility and the observed trends in excess oxygen decrease in the upper ocean.

Due to the relationship between excess temperature and oxygen change, the most substantial excess temperature change over each time period also occurs at depths between 250-300 meters, reaching a change of 0.35°C over the 1992-2015 period. Below 1000 meters, the magnitude of the excess temperature change generally remains less than ±0.05°C and hence drives very little deoxygenation.

DIC change is dominated by the excess term in the upper 1000 m (Fig. 5g), due to increasing anthropogenic CO$_2$ levels. Trends in excess DIC changes mirror those in excess temperature since they are linearly related via the coefficient $\alpha$, which is constant in time. Excess DIC reaches a maximum change of 21.5 $\mu$mol kg$^{-1}$, occurring between 1992 and 2015 at a depth




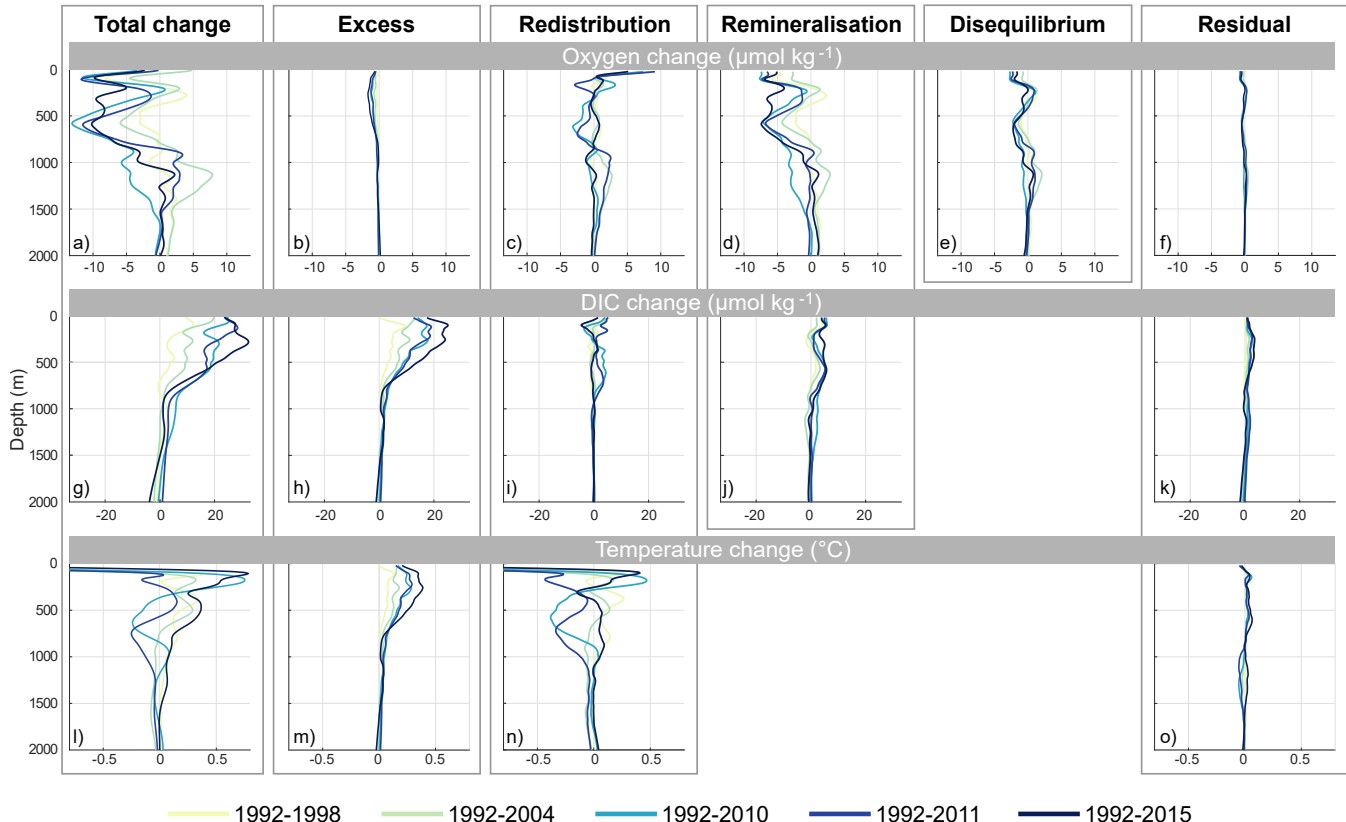

**Figure 5.** The oxygen ($\mu$mol kg$^{-1}$), DIC ($\mu$mol kg$^{-1}$), and potential temperature (°C) decomposition results averaged over the A05 section, from 30-70°W: a) the total temperature change since 1992, b) the excess change, and c) the redistributed change. d) shows the residual in the method, i.e. the difference between the total potential temperature change and the sum of the changes attributed to each process.

of ∼270 m (Fig. 5h). Due to the ratio of oxygen consumption and DIC production during remineralisation, we also see a clear temporal trend in remineralisation DIC change, with increasing production over time. Remineralisation change peaks at a depth of ∼600 m, reaching around 5.7 $\mu$mol kg$^{-1}$. At lower depths, DIC changes becomes close to zero and in some cases slightly negative, but this negative change is generally smaller than the internal accuracy of the GLODAP data product. As with

temperature, no distinct temporal patterns are discernible in redistributed DIC changes. The decomposition shows that while the anthropogenic increase in CO$_2$ fluxes dominate, remineralisation changes are also responsible for a significant amount of the increased DIC in the upper ocean. Excess DIC change in the section is strongly correlated with the total DIC change ($r^2$ = 0.88), but the correlation between the remineralisation change and the total change is also high ($r^2$ = 0.62).





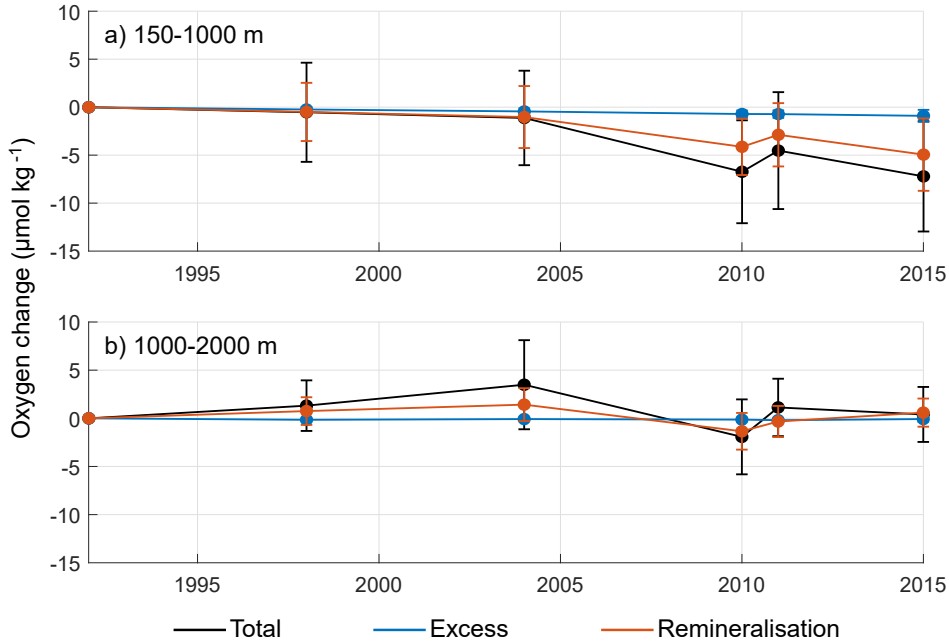

**Figure 6.** Oxygen change over time ($\mu$mol kg$^{-1}$), averaged over the A05 section (30-70°W), and integrated over a) the upper 1000 m, and b) 1000-2000 m. The total change is shown in black along with the terms that show the strongest trends: excess change (blue) and remineralisation (red), along with the standard deviation in each.

## 4 Discussion

Although we only observe one section within the North Atlantic, we find that alterations in remineralisation are responsible for up to half of the total oxygen decrease seen in the upper 2000 m during recent decades. By linking changes in DIC and oxygen, we ensure that any process that impacts remineralisation is taken into account. While we don't consider the impact of changes in stoichiometry, and instead assume that the stoichiometric ratio applied remains constant in time and space, using a constant ratio does simplify the method and allow for the same technique to be used to investigate deoxygenation globally. In

reality, there will be variations from the Redfield ratio within our considered section. However, altering the coefficient linking remineralisation changes in oxygen and DIC by as much as $\pm$ 50% has only a very minor impact on the average change due to each process (Fig. B1).

Previous studies have suggested that with increased ocean warming, remineralisation tends to occur at shallower depths (Oschlies, 2021), which may play a role in driving the increased remineralisation in the upper 1000 m. Here, we calculate

the oxygen change due to each process at each longitude-depth point individually, so the observed change in remineralisation processes could be a result of either an increase in the total amount of production exported and remineralised, a change in the spatial patterns of remineralisation, or both. This method alone does not allow us to determine which is driving the changes within the A05 section, and further work is required to evaluate the drivers responsible for the increased oxygen consumption



via remineralisation. However, there has been little change in production in the surface of the subtropical North Atlantic over
the same time period (Macovei et al., 2019). If production has not significantly increased at the surface, thus leading to higher
rates of export to depth, and higher rates of remineralisation, it instead suggests that it is changes in ocean physics that are
driving the increased remineralisation. This theory is also supported by previous studies that have shown that more carbon is
being stored in the ocean interior despite export production to depth decreasing (Wilson et al., 2022). However, discerning the
exact physical processes responsible for these changes is complicated and warrants further study.

While remineralisation changes dominate deoxygenation, temperature and DIC change over this period are dominated by
different mechanisms. The temperature change in the A05 section is driven primarily by redistribution in the upper 2000 m, in
agreement with previous studies (Bindoff and Mcdougall, 1994; Desbruyères et al., 2017), and excess change is the dominant
driver of increasing DIC in the upper 2000 m due to the influence of increasing anthropogenic atmospheric $CO_2$. Remineralisation changes are however also responsible for a small but significant amount of the increased DIC in the upper ocean. While
excess temperature and oxygen changes are smaller in comparison to the effect of other drivers, excess temperature change
nearly trebled over the period of the A05 section occupations and is now of a similar magnitude to redistribution within the
upper 500 m. Given the magnitude of this increase, it could be expected that this term will become dominant in the surface
waters in the near future, as suggested by previous studies (Bronselaer and Zanna, 2020; Zika et al., 2021). Since excess oxygen change is driven by temperature-driven solubility change, further surface warming will lead to increased excess-driven
deoxygenation. The excess oxygen change has already doubled in magnitude between 1992 and 2015. The average total oxygen change between 1992 and 2015 was -7.21 $\mu$mol kg$^{-1}$ in the upper 1000 m. For the excess oxygen change to reach this
magnitude, an excess temperature change of 0.73 °C would be required.

Apparent Oxygen Utilisation (AOU) is often used as a measure of the oxygen consumed by remineralisation in the ocean
interior. While this is similar to the remineralisation change term that we calculate here, there are several important differences.
Our method calculates the remineralisation rate, whereas AOU calculates the absolute consumption of oxygen. AOU also
assumes there is no disequilibrium at the ocean surface when oxygen is first absorbed from the atmosphere, instead combining
the disequilibrium and remineralisation components that we resolve separately in this method. The final and most important
difference is the way that temperature change and its impact on oxygen change is interpreted. Here, we separate the temperature
change due to perturbations in air-sea heat flux, where ocean warming is associated with a reduction in ocean solubility and
therefore deoxygenation, i.e. the excess temperature and oxygen changes. In contrast, in the upper ∼1000 m a redistributed
warming of the ocean (or a downward motion of an isotherm) will result in an increase in oxygen concentration, the result
of a positive correlation between temperature and oxygen above 8°C at the A05 section. While we separate these terms,
if these redistributed temperature changes are not correctly accounted for separately, they will be incorrectly accounted for
in the remineralisation component. For instance, if the total temperature changes that we observe are all interpreted as a
warming signal associated with air-sea fluxes, this will overestimate the warming driven de-oxygenation, and underestimate
the consumption via remineralisation. Therefore, the change in AOU will in this case be much smaller than, and not comparable
to, the change in remineralisation calculated using this method. AOU will also mask the systematic signals that we observe
over time by introducing the large variability seen in the redistributed signal.





## 5  Conclusions

We used a new technique linking oxygen, DIC and temperature changes, to show that remineralisation dominates the de-oxygenation that has occurred in recent years along the A05 transect in the subtropical North Atlantic. Spatial patterns in remineralisation change closely matching those in the total oxygen change within the upper 2000 m. Further work is needed to understand what is driving this increase in remineralisation oxygen consumption, whether it is occurring in other regions, and whether this method gives comparable results when applied to model output, to determine if this could be a potential cause for

the mismatch between observed and modelled deoxygenation in the global ocean.





| Cruise dates | Number of stations | GLODAP expocode | GO-SHIP expocode | Notes |
|---|---|---|---|---|
| Jul-Aug 1992 | 112 | 29HE19920714 | 29HE19920714 | |
| Jan-Feb 1998 | 130 | 33RO19980123 | 31RBOACES24N_2 | |
| Apr-May 2004 | 125 | 74DI20040404 | 74DI20040404 | |
| Jan-Feb 2010 | 135 | 74DI20100106 | 74DI20100106 | |
| Jan-Mar 2011 | 167 | 29AH20110128 | 29AH20110128 | DIC calculated from pH |
| Dec-Jan 2015 | 145 | 74EQ20151206 | 74EQ20151206 | -4 $\mu$mol kg$^{-1}$ DIC adjustment wrongly applied in GLODAPv2.2002 removed (removed in GLODAPv2.2023) |

**Table A1.** Additional information about the six cruises in the A05 section between 1992 and 2015.

## Appendix B: Derivation of System of Oxygen Change Equations

The total change in oxygen ($\Delta O$), temperature ($\Delta T$) and DIC ($\Delta C$) can be defined as follows, where the subscripts *e*, *rd*, *rem* and *d* describe the excess, redistributed, remineralisation and disequilibrium change, respectively.

$$\Delta O = \Delta O_e + \Delta O_{rd} + \Delta O_{rem} + O_d, \tag{B1}$$

$$\Delta T = \Delta T_e + \Delta T_{rd}, \tag{B2}$$

$$\Delta C = \Delta C_e + \Delta C_{rd} + \Delta C_{rem}, \tag{B3}$$

The excess change in temperature can be linked to both the excess change in oxygen and in DIC. Excess changes in oxygen are driven by excess changes in temperature via changes in solubility, described by the coefficient $\eta$:

$$\Delta T_e = \eta \Delta O_e \tag{B4}$$





Excess changes in DIC and temperature are linear related via a coefficient $\alpha$, due to increasing anthropogenic $CO_2$:

$$\Delta T_e = \alpha \Delta C_e \tag{B5}$$

Redistributed changes in both oxygen and DIC can be related to redistributed changes in temperature via the coefficients $\beta_O$ and $\beta_C$ respectively due to the effect of local circulation on each property:

$$\Delta T_{rd} = \beta_C \Delta C_{rd}, \qquad\qquad\qquad \Delta T_{rd} = \beta_O \Delta O_{rd}. \tag{B6}$$

Finally, the remineralisation changes can be linked via a coefficient $\gamma$, which describes the stoichiometric ratio between DIC and oxygen:

$$\Delta C_{rem} = \gamma \Delta O_{rem} \tag{B7}$$

Rearranging these equations, means that the total change in temperature can therefore be defined via changes in oxygen:

$$\Delta T = \Delta T_e + \Delta T_{rd} \tag{B8}$$

$$= \eta \Delta O_e + \beta_O \Delta O_{rd} \tag{B9}$$

The total change in DIC can then also be computed via changes in oxygen:

$$\Delta C = \Delta C_e + \Delta C_{rd} + \Delta C_{rem} \tag{B10}$$

$$= \frac{\Delta T_e}{\alpha} + \frac{\Delta T_{rd}}{\beta_C} + \gamma \Delta O_{rem} \tag{B11}$$

$$= \frac{\eta}{\alpha} \Delta O_e + \frac{\beta_o}{\beta_c} \Delta O_{rd} + \gamma \Delta O_{rem}. \tag{B12}$$

The total oxygen, DIC and temperature can now be defined in terms of changes in oxygen terms and the coefficients $\alpha$, $\beta_C$, $\beta_O$, $\eta$ and $\gamma$, and can be written as the following matrix equation in the form $Ax = b$:

$$\underbrace{\begin{bmatrix} \eta & \beta_o & 0 & 0 \\ \frac{\eta}{\alpha} & \frac{\beta_o}{\beta_c} & \gamma & 0 \\ 1 & 1 & 1 & 1 \end{bmatrix}}_{A} \underbrace{\begin{bmatrix} \Delta O_e \\ \Delta O_{rd} \\ \Delta O_{rem} \\ \Delta O_d \end{bmatrix}}_{x} = \underbrace{\begin{bmatrix} \Delta T \\ \Delta C \\ \Delta O \end{bmatrix}}_{b}. \tag{B13}$$

This matrix equation can then be rearranged in order to solve to find the unknown values in $x$:



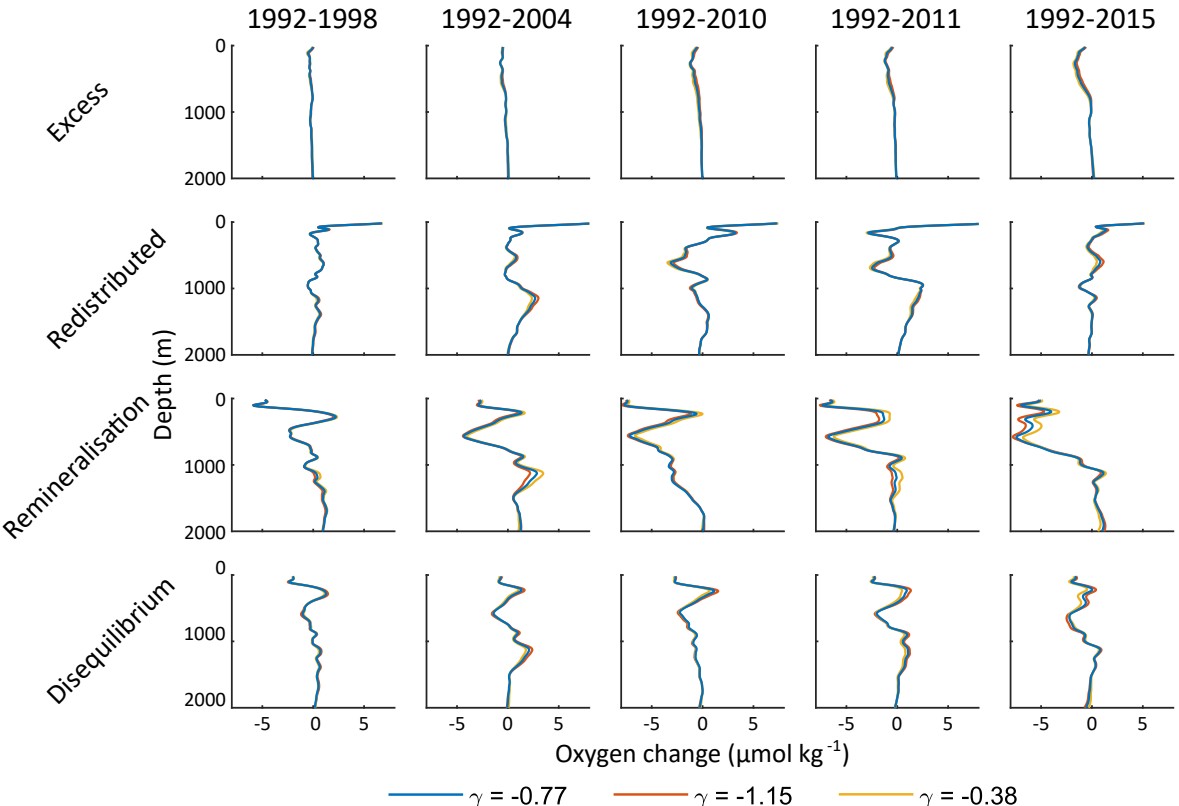

**Figure B1.** The sensitivity of the oxygen change decomposition to changes in the coefficient $\gamma$, which links remineralisation changes in oxygen and DIC. Each row shows the oxygen change due to a different process for the different years computed using a constant $\gamma = -0.77$ as used in this study (blue), computed with a $\gamma$ value 50% lower (orange) and 50% higher (yellow).

$$\underbrace{\begin{bmatrix} \Delta O_e \\ \Delta O_{rd} \\ \Delta O_{rem} \\ \Delta O_d \end{bmatrix}}_{x} = \underbrace{\begin{bmatrix} \eta & \beta_o & 0 & 0 \\ \frac{\eta}{\alpha} & \frac{\beta_o}{\beta_c} & \gamma & 0 \\ 1 & 1 & 1 & 1 \end{bmatrix}}_{A^{-1}} \underbrace{\begin{bmatrix} \Delta T \\ \Delta C \\ \Delta O \end{bmatrix}}_{b}. \tag{B14}$$



*Code and data availability.*  The GLODAPv2.2002 data is publicly available at https://glodap.info and is described by Lauvset et al. (2022). The GO-SHIP Easy Ocean gridded hydrographic data is available at https://zenodo.org/records/13315689 and is described by Katsumata et al. (2022). The package used to interpolate the GLODAP data onto the GO-SHIP sections was developed by Charles Turner and can be found at https://github.com/charles-turner-1/GLODAP_Section_Gridder.jl.

.

*Author contributions.*  RNCS completed the analysis, wrote the draft and made the figures, ELM proposed the study, SKL provided advice on the data, CET provided the data processing methods, TWNH provided advice on the methods, and ELM, SKL NG, RS provided advice on the analysis.

*Competing interests.*  None

*Acknowledgements.*  RNCS, ELM and NG would like to acknowledge funding from the strategic project "The breathing Ocean" from the
Bjerknes Centre for Climate Research.



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
