# Peer review of "Remineralisation changes dominate oxygen variability in the North Atlantic"

_EGUsphere, 2025_

## Author Comment (AC1)

We thank the reviewer for their helpful and constructive comments. The comments are listed below in black, with our response to each in blue. We hope that you find the improved manuscript suitable for publication.

The manuscript "Remineralisation changes dominate oxygen variability in the North Atlantic" uses data from repeat hydrographic occupations of the A05 line in the to investigate long-term oxygen changes North Atlantic. They partition the change into four different terms: the excess, redistributed, remineralization, and disequilibrium oxygen change, reflecting different underlying processes, and find that the remineralization term is responsible for most of the change from 1992 to 2015.

I think overall this paper is well written and the methods are sound, but I think some improvements could be made. In particular, I think the role of ocean circulation in the observed changes should be highlighted more, and there are some inconsistencies throughout the text, as well as some parts of the methods section that could be explained better

**General comments**

Reading the manuscript, my first impression from the title and much of the text was that the authors found that the rate of respiration in the North Atlantic increased since 1992 ("Remineralisation changes dominate"). But as stated in line 269-272 and in the abstract in line 15-17, it actually is likely due to circulation changes, i.e. changes in "accumulated" upstream remineralization, rather than local ones. I think some more explicit mention of the role of circulation would make this clearer.

Below are some ideas for changes that could help make this point; I don't think all of these need to be included, but implementing at least some of them would help highlight this point:

- Change the title and/or text to use something more reflective of the role of circulation in the remineralization signal; e.g. "upstream remineralization" or "accumulated remineralization"

  Thank you for the suggestion. Since our technique alone does not show that circulation is driving the remineralisation change, and we can only speculate that this is the case, we have kept the title the same. However, we have added extra information throughout the manuscript to explain the different potential drivers of the change in remineralisation, and why we believe circulation to be the most likely driver.

- Move the sentence in line 10-12 ("This remineralization-driven change…") up to earlier in the abstract, e.g. to line 5 after the sentence "Mechanisms responsible for…" (might need to rewrite a bit)

  This has been moved earlier and reworded to: "Mechanisms responsible for this deoxygenation include (i) solubility-driven deoxygenation due to ocean warming, and (ii) changes in the remineralised signal due to either a change in the supply of biological material to depth or a change in circulation leading to change in the residence time of water, and hence the accumulation of the remineralised oxygen deficit, or a combination of both." (Lines 3-6)

- In lines 35-40, explicitly mention ocean circulation. I believe there are several studies linking the "indirect" (not solubility related) oxygen loss to circulation/ventilation changes, such as Oschlies et al 2018 or Schmidtko et al 2017, and this explains part of the model-obs discrepancy (Buchanan & Tagliabue 2021).

  Thank you for the references, the following sentences have now been amended:

  "The remaining warming has been linked to changes in ventilation and circulation, which can lead to changes in the residence time of the water, allowing for more remineralisation to occur (Oschlies et al., 2018, Schmidtko et al., 2017)." (Lines 35-36)

  "It has been hypothesised that the disparity between observational and modelled oxygen decline is due to uncertainty in either historic atmospheric forcing data, the representation of biogeochemical processes in models (Oschlies et al., 2017), or coarse resolution models not accurately representing changes in circulation and ventilation (Buchanan & Tagliabue, 2021)". (Lines 46-48).

- It could also be useful to move parts of the following paragraph (lines 41-49) up. Currently, you discuss drivers of ocean deoxygenation in line 32-36, then talk about the model-observation disparity in line 36-40, then go back to discussing drivers of deoxygenation more generally in line 41-49, so rearranging this could improve the flow

  This has now been rearranged.

- In section 2.2, explicitly mention circulation changes. Line 83-85 talks about "export of organic matter" and "spatial patterns of remineralization"; I think maybe the latter is meant to refer to circulation, but it would be good to spell this out more clearly. And in the same section, circulation is mentioned in line 76-77 as part of the redistribution term. Does this refer to local circulation versus global/upstream circulation for the remineralization term? If so it might be useful to state that explicitly

  This section has now been edited to explicitly state that redistribution refers to local circulation changes, while change in remineralisation can be due to changes in largescale circulation:

  "The vertical redistribution of the oxygen already in the water column is a result of local circulation changes, or reorganization of the local oxygen structure" (Lines 83-85)

  "Changes in remineralisation can be either due to a change in the rate of export of organic matter leading to an altered rate of remineralisation at depth, or to changes in the spatial patterns of remineralisation, i.e. because changes in large scale circulation patterns affect the residence time of water and hence the accumulation of the remineralised oxygen deficit." (Lines 91-93)

- The beginning of the discussion (line 255-256) is another opportunity to more explicitly state that the "alterations in remineralization" could be either local or upstream

  This has now been added: "Based on our new methodology and analysis of one repeat hydrographic section in the North Atlantic, we find that alterations in remineralisation, either local or upstream, are responsible for…" (Lines 278-279)

- While I agree that "discerning the physical processes responsible is complicated" (line 273-274) I still think the discussion could mention some studies on physical transport for context. E.g. could part of the oxygen trend be explained by the decreased flow of AAIW mentioned in line 60 (Hernandez-Guerra et al 2014) or results from the RAPID array (Smeed et al 2018; Johns et al 2023) which showed a decrease in AMOC strength around 2008-2010?

  More discussion has now been added about the possible role of AMOC:

  "Further work beyond the decomposition method is required to determine the drivers responsible for the increased oxygen consumption via remineralisation with certainty. A significant increase in primary production would lead to higher rates of export to depth, and higher rates of remineralisation. However, there has been little change in primary production in the surface of the subtropical North Atlantic over the same time period (Macovei et al., 2019). This suggests that either there is has been change in the export ratio, with less of the productivity exported to depth, or that changes in large-scale circulation are driving the increased remineralisation. Modelling studies have also suggested that in future more carbon will likely be stored in the ocean interior, despite export production to depth decreasing as a result of a slowdown in ocean circulation (Wilson et al., 2022). This gives further credibility to the hypothesis that changes in circulation could be driving the increased remineralisation along the A05 section. Large-scale circulation changes at the A05-Section are strongly influenced by changes in Atlantic Meridional Overturning Circulation (AMOC). The strength of the AMOC at the latitude of A05 has been monitored since 2004, and weakened between 2004 and 2010 (Smeed et al., 2014; Johns et al., 2023). This decline may have led to increased residence times, allowing for more remineralisation to occur before the water reaches the A05 section. This would also explain why the dominant decreases in remineralisation are visible after 2004 and why they occur within the upper 1000m and in the western part of the Section. However, discerning the exact physical processes responsible for these changes is complicated and warrants further study." (Lines 290-304).

**Specific comments**

Line 4: rephrase "solubility-**driven** deoxygenation **driven by**…"

This has been rephrased to "solubility-driven deoxygenation due to ocean warming" (Line 4)

Line 29: "oxygen at the surface is close to saturation" – this isn't true in many deep/mode water formation regions like the North Atlantic (e.g. Clarke & Coote 1988, Wolf et al 2018)

and Southern Ocean (Bushinsky & Cerovecki, 2022), and also seems inconsistent with having a "disequilibrium" term

This sentence has now been removed.

Line 71: I assume the change is actually calculated as 1998 minus 1992 (etc), so the way it's phrased currently as "1992-1998, 1992-2004, and so forth" could be confusing

We agree this could be confusing, since the change is calculated as 1998 minus 1992. We have now changed this in both the text and figures to 1992 to 1998 etc.

Line 76: Does "globally" here refer to the actual globe, or just globally within the dataset, i.e. along the section?

This refers to the actual globe.

Line 85: There's something missing in the phrase "The final, disequilibrium change.." – should it be "The final **term**"?

Thank you, this has now been corrected to say "The final term, disequilibrium change…" (Line 95)

Line 100: Should there be a Delta before O_d?

This should have included a Delta and has now been corrected.

Line 112/eq. 4: It would be good to already briefly describe the coefficients here so the reader doesn't have to search for them through all the sections. The equation could also be moved down to be between section 2.6 and 2.7, so that the individual equations are described first, and the system of equations is then formulated once all the terms have been defined

The following paragraph has been added below Equation 4: "The coefficient η links changes in excess oxygen to those in excess temperature, via a change in solubility. The redistribution coefficients $β_O$ and $β_C$ link changes in oxygen and DIC redistribution respectively, to temperature redistribution. α describes the linear relationship between excess DIC increase and excess warming, which is dominated by anthropogenic changes, and γ describes the approximate ratio of oxygen and DIC change during remineralisation. The calculation of these coefficients is described below, and the full derivation of Equation 4 is shown in the Appendix." (Lines 126-130)

Line 122: Does the "gradient of the temperature-oxygen solubility curve" mean that you take a T-O plot with contours of solubility at 100% saturation, then use that to determine the solubility gradient as a function of T and O? That could be good to show graphically in addition to (or instead of) Fig. 3b showing the values of the coefficient

Thank you for the suggestion. Figure 3b has been changed to show the change in oxygen with temperature due to solubility change, when initial temperature is defined as the mean, minimum and maximum initial temperature that occurred within the 1992 A05 section. The solubility coefficient η, is then computed as the gradient of these curves, given that they are close to linear over the temperature changes seen within the A05 section. This has also been described in the figure caption.

Line 140: Is the redistribution term just vertical redistribution for each profile? Since the equations only include vertical gradients I assume that's the case, but if so it should be noted explicitly here and/or earlier on (e.g. line 75)

Yes, the redistribution change is only vertical. This has now been clarified in the text here, as well as when the redistribution term is first introduced:

"The vertical redistribution of the oxygen already in the water column is a result of local circulation changes, or reorganization of the local oxygen structure" (Lines 83-84)

and

"The redistribution coefficients $\beta_C$ and $\beta_O$ relate vertical redistribution driven changes in temperature to redistribution driven changes in DIC and oxygen, at a given geographical location." (Lines 156-157).

Line 148-149: Does the shift from negative to positive coefficients correlate with any particular water masses?

The change in sign does not appear to occur at the boundary of a water mass, and instead occurs within the AAIW layer (see figure below). This has now been mentioned in the results section: "The lines of near-zero redistribution close to the surface and at a depth of around 800 m (Fig. 4k-o) are simply due to the change in sign of the $\beta_O$ coefficient as it switches from positive to negative, and have no correlation with any particular water masses". (Lines 245-247)

[Figure]

Line 153/eq.10: Add line break between C and O equations to be consistent with other eq.s

This has now been changed.

Line 163-164: Since weights are used in eq. 4 it is no longer just defined as Ax = b as stated in line 109, correct? It would be useful to restate the form the equation takes with the weight matrix, but could do so inline instead of as a numbered equation

Equation 4 has been rewritten in the form W1 A x = W2 b, where W1 and W2 are weights used to regularise the system due to it being under-determined.

Line 191-192: Why was the region from 30-70W chosen? Also it would be good to show the box used for the averaging in at least one panel of fig. 4, and/or highlight it in fig 1.a

A box has now been added to the map in Fig. 1a to show the longitude range averaged over in Figs 5 and 6.

The region was chosen to exclude the areas in the far east and west of the transect, where the total oxygen change at the surface is very different to the rest of the section, with strong increases in some years. This is likely due to the effect of Mediterranean water in the east, and the influence of boundary currents in the west. Within this central region, the variability in the upper 1000 m is generally more spatially robust, and so while it would be interesting to further investigate the changes in the east and west of the section, in this study we considered it more logical to look at the central region. The reasons for choosing the region to analyse have been further explained in the manuscript.

"To better understand the drivers of the average property changes across the A05 section, we focus on the region between 30–70°W (black box in Fig. 1a). This allows us to focus on the central region that has more robust trends in the upper 1000 m, with clear deoxygenation over time, while removing the impact of regional oxygen increases occurring in the eastern and western boundaries of the section." (Lines 212-215)

Line 200-202: Can the changes over different depth ranges be related to any particular water masses? E.g. subtropical mode waters or Antarctic Intermediate Water. Looking at fig. 1 it sems that there is a clear oxygen minimum/DIC maximum around 800m, which could be indicative of AAIW, and this is near where the changes are most pronounced

Isopycnals depicting the different water masses in the section have been added to Fig. 4, as well as to the mean A05 sections in Fig. 1. Throughout the results section, we have now referred to the water masses that change is occurring in, with the deoxygenation primarily occurring within the North Atlantic Central Water, and very little change in the North Atlantic Deep Water.

Line 207-208: The last clause seems a bit redundant; "**with consumption of oxygen via remineralisation generally increasing** over time in the upper 1000 m, **indicating an increase in the total amount of remineralisation.**"

This has now been removed.

Line 215: Rephrase "Excess increase peaks"  - actually a decrease

This has been rephrased to "The largest excess changes occur at…" (Line 240)

Line 219: "around 1000m" – change to "around 800m" to be consistent with line 149?

This has now been changed (Line 244)

Line 235/line 140: How is temperature redistribution defined? Is it just the redistributed oxygen change times the coefficient (eq. 10)?

Yes, the temperature terms are computed by adding the oxygen terms back into the equations used to compute the coefficients. The following line has now been added to explain this:

"Once the redistributed oxygen term is calculated via Equation 4, it can be used to compute the temperature and DIC redistribution via Equations 10 and 11." (Lines 172-173)

Fig. 4: To provide more context for the changes, it would be helpful to show mean sections of T, O2, and DIC either here or in Fig. 1.

Mean sections of temperature, DIC and oxygen have now been added as Fig. 1e-g

For panels f)-j), I think it would be better to have the same color bar range as the other rows. Alternatively, you could keep the same range but a slightly different color map, e.g. purple-orange instead of blue-red.

The colour map for this panel has now been changed to purple-orange to make it clearer that there is a different colour scale to the other panels.

In the caption, I'm not sure what "the surface 150m is omitted from a)" means, since the panels all seem to go to 0 on the y axis, and I'm guessing it also wouldn't be just for one of the 25 panels

Sorry, this was added to the wrong caption – it should have been included in the caption for Fig. 6 and has now been moved.

Line 239: "Due to the relationship between excess temperature and oxygen change" – this makes it sound like the oxygen changes are driving temperature changes; this may be true for the way it's calculated here, but mechanistically it would be more correct the opposite way, i.e. excess oxygen changes are caused by the excess temperature change. I think you could just mention this in the discussion of excess oxygen change (line 213) instead

This has been moved up to the discussion of excess oxygen change as suggested.

Fig. 5: The colors aren't quite consistent with Fig. 1 since the color scheme is the same but with 5 values instead of 6. As a result, e.g. the 1992-2004 line in Fig. 5 is essentially the same color as the 1998 line in Fig. 1. I suggest using the colors corresponding to the "end year" in F1 in F5 so that for example the 1992-1998 line in this figure and 1998 in fig. 1 are the same color

Using the colours from the end year made it difficult to see the difference between them, since there was less contrast between the first and last colour. Instead, we have changed the colours in Figure 1 to be completely unique, and so not comparable with the colours in other figures.

Line 256: It would be good to be more consistent with the depth ranges discussed in the text. In section 3.2, changes are discussed for 150-500m, 500-1000m and 1000-200m in the first paragraph (line ~200, table 2), but later separated into 150-1000m and 1000-2000m (Fig. 6/line 228-233). Then this paragraph goes on to talk about "the upper 2000m", before the next one talks about "the upper 1000m".

We have changed some of these (such as the calculation in the comment below), but generally when the results are for say 150-1000 m, it's because the trend is observed in both the 150-500 m and 500-1000 m layers. Therefore, we have kept some of them the same to avoid having to write out both depth ranges each time.

Line 287: "an excess temperature change of 0.73C would be required" – you could state here how that relates to the observed change in fig. 5, i.e. about 3-4x (?) the warming observed so far. Again this would be easier compare if table 2 would use the same depth range as used in the text

We have now done the calculation for the 150-500 m layer instead to fix with the text, and the value has been compared to the actual excess temperature change during this period:

"The average total oxygen change between 1992 and 2015 was -7.82 µmol kg$^{-1}$ in the upper 150-500 m layer. For the excess oxygen change to reach this magnitude, an excess temperature change of 1.74°C would be required, assuming the average value for $\eta$ at these depths. This is around five times greater than the actual average excess change during this period." (Lines 316-319)

Line 298: Repetitive sentence – "**are not correctly accounted for** separately, **they will be incorrectly accounted for** in the remineralisation component."

This has been changed to "While we separate these terms, if these redistributed temperature changes are not correctly accounted for separately, they will instead be wrongly included in the remineralisation component." (Lines 337-338)

Line 307: "matching" -> "match"

Thank you, this has been corrected

Line 308: Some extra words? "this increase in **remineralization oxygen consumption"**

This has now been changed to "remineralisation-driven oxygen consumption"

---

## Author Comment (AC2)

We thank the reviewer for their helpful and constructive comments. The comments are listed below in black, with our response to each in blue. We hope that you find the improved manuscript suitable for publication.

In this study, the authors use ship-based observations along a repeat section in the North Atlantic to disentangle the mechanisms driving ocean deoxygenation, i.e., solubility and remineralization changes, and changes in ocean circulation. By using a matrix of equations, the authors have decomposed the changes in oxygen, DIC and temperature over the past nearly three decades. This study shows that the deoxygenation in upper 2000m is dominated by an increase in remineralisation via circulation changes, and the increasing temperature is becoming a more and more important mechanism to drive the upper ocean oxygen loss. The main conclusion is straight forward and the manuscript is generally well written.

We have some general comments here:

1. The writing structure of the results part is a bit fragmentary, especially in section 3.2. I encourage the authors to regroup the current paragraphs (sentences).

Section 3.2 has been edited to improve the flow of the text.

2. It is a good study to show how the interior ocean deoxygenized, through multiple GOSHIP transects, but we have a general impression that this study is a bit incomplete. For example, (1) the authors didn't explain well how they deal with the potential bias between the several ship tracks, which might result from the different measurement methods, different seasons of the ship track undertaken; (2) there is a clear different pattern of the oxygen changes between the west and east parts of the transect. It's not clear why the authors didn't discuss this signal, but instead, they took the average of the whole section. Please refer to the more specific comments below.

Using the GLODAP data product rather than raw bottle measurements resolves much of this potential bias. Extensive quality control has already been undertaken to produce the GLODAP dataset, and adjustments applied when required. The adjustments are applied in order to remove the biases that arise due to differences and errors in measurement and calibration techniques. We have also consulted with the GLODAP team, and there were no major changes in methods used for measuring oxygen or DIC data between these cruises. All the data used in this analysis was collected and analyzed using standard operating procedures, outlined in the GO-SHIP manuals (Langdon, C., 2010 and Dickson et al., 2007). More information about the GLODAP product has now been added to the manuscript:

"The GLODAP data product has undergone extensive quality control, with adjustments applied when required, in order to remove any biases between cruises (Olsen et al., 2016)." (Lines 67-68)

"The data collected during these cruises were collected and analysed using standard operating procedures as outlined in the GO-SHIP manuals (Langdon et al., 2010, Dickson et al., 2007)." (Table A1 caption)

While there are differences between the east and west of the section, we find it more logical to focus on the central region of the section, where the most robust deoxygenation trends occur, and are not influenced by the extreme localised values observed in the most westerly

and easterly parts of the section. If we average the changes in oxygen over the eastern and western halves of the section, there would still be significant regional differences within the halves, particularly due to the influence of Mediterranean water in the east and boundary currents and Florida Strait in the West. Therefore, while it would be interesting to use the same method to look at more localised changes, we believe it is sensible to first focus on the more robust changes in the average of the central region.

To make this consistent throughout the manuscript, we have instead included the average temperature, DIC and oxygen profiles for the whole of the central region of A05 in Fig. 1, so they are averaged over the same region as in Fig. 5. We have also included the mean properties of the section in Fig. 1e-g so spatial variability is still included. We have included further discussion about why we choose to average over this section:

"we focus on the region between 30–70°W (black box in Fig. 1a). This allows us to focus on the central region that has more robust trends in the upper 1000 m, with clear deoxygenation over time, while removing the impact of regional oxygen increases occurring in the eastern and western boundaries of the section." (Lines 212-215)

3. The title suggests that remineralization is the primary cause of oxygen change, but the abstract and the main text indicate that it is actually circulation changes that lead to a change in the spatial pattern of remineralization. Please reword the title to make this clearer

Thank you for the suggestion. Since further work is required to be sure that circulation driving the remineralisation change, and we can only speculate that this is the case with this method, we have kept the title the same. However, we have added extra information throughout the manuscript to explain the different potential drivers of the change in remineralisation, and why we believe circulation to be the most likely driver.

Overall, agreements between treatments of independent ship transects and justification of different signal between west and east seem to be robust. The manuscript is also readable and well organized. However, before publication, several methodological problems need to be fixed (declared), discussions need to be refocused, and some typos and visualization issues need to be fixed.

Method.2.1 We think the important information about how these GO-SHIP transects deal with the samples is missing. Were there any changes in methodology over the three decade time period? Although the authors listed the expocode in Table A1, we suggest adding more information in the method section, such as a simple, summarized protocol (consistent part for all the transects) and a transparent reminder of any potential biases and differences from transect to transect.

The GLODAP data product already takes into account any changes in methodology, by applying adjustments to the cruise data if required to correct any biases. However, we have consulted with the GLODAP team, and have been told that there were no major changes in the methods used for sampling and processing oxygen and DIC data between these cruises. All the data used was collected and analyzed using standard operating procedures, as outlined in the GO-SHIP manuals (Langdon, C., 2010 and Dickson et al., 2007). A line has now been added in the caption of Table A1 to state this, and the manuals cited.

Line 83: 'However, the water mass properties we are interested in change more slowly, and so are resolved by the ~5-year frequency.' Do the authors have any reference for this?

This sentence has now been rephrased to explain that we are interested in those longer-term changes that are resolved by the frequency of cruise:

"However, we are interested in the longer-term changes in water mass properties, which are resolved by the ~5 year frequency."

Figure 1.

I would put the profiles from the west above (i.e., b-d) the profiles from the east (i.e., e-g). Since we read from left to right, this is how I would expect the outline of the figure.

The profiles for the east and west of the section have now been removed. Instead, we show the profiles averaged over the central part of the A05 section, to match the profiles in Fig. 4. However, the temporal average of temperature, oxygen and DIC for the entire section has now also been included, so the spatial variability can be seen across the section.

And we would suggest the authors use another way to plot the ship-transects in the map (1a). I understand that the authors have stated they interpolated the six transects to a uniform shape, but a black line with a part stack on the continent is not very informative. Perhaps showing the location of the casts would be helpful as a piece of information to aid the readers in understanding the transects.

Thank you for the suggestion, the line on the map has now been replaced by markers showing the individual CTD casts of the A05 cruise in 1992. As it is difficult to display the stations from every year, due to many of them overlapping, the 1992 station locations are shown since all changes in oxygen are calculated as the change from 1992.

What's the purpose of separating into two sections (west/east)? It seems like the whole study discussed the transect as a whole (mainly focused on 30-70W).

The authors switch between using the 30–70W average and the whole transect in Figures 1 – 5. If there are important differences between the west and east section, they should be presented and discussed separately and if there are no differences, then this should be stated up front and thereafter only the 30-70W averages shown and discussed.

Thank you for the suggestion. We agree that it was not consistent to split the section into east and west in Fig. 1 when we do not separately analyse the two halves of the section. Instead, we have changed Fig 1 to have only the averaged profiles across the whole of the A05 section.

Figure 2. Since this figure is discussed before the acronyms are explained in the main text, I think it would help the reader to explain what e, d, rd, and rem stand for in the figure caption. Or maybe include the acronyms at the beginning of section 2.2.

These acronyms have now been defined in the caption of Fig. 2.

Lines 100 – 102: the text preceding the equations goes through them in order of temperature, DIC, and O2. I would suggest listing the equations in that order for ease of reading. Equation 10 has two equations; please numerate these separately.

The order of equations has now been changed, and Equation 10 has been split into two separate equations.

Section 2.6: Would a different Refield Ratio change the result? I see this is discussed later, so I suggest bringing this up at this point.

This is included in the discussion, but the following has now been included earlier in Section 2.6: "While variation in the Redfield ratio was tested, even changing the value by as much as 50% has little impact on the overall findings (Fig. B1)." (Lines 181-182)

Line 211 & 252-253: could the authors please also report p values?

p values have now been added for all calculations of $r^2$.

Line 248: Suggest rephrasing "lower depths" to deeper depths, as I assume this is what was meant? Lower is ambiguous in this context.

This has now been changed to deeper depths.

Line 236. Steadily increasing trend? Over time?

This has been changed to "steadily increasing trend over time". (Line 262).

Line 240. Could the authors please refer to a plot or table?

Fig. 5m has now been referenced. (Line 244)

Figure 6. Sometimes we found it hard to see the error bars since they overlap, and the straight line between each dot makes it look like a linear trend, although it's not. Thus, we suggest using a better way to visualize this figure, for example a bar chart with stacked components.

We have tried to reproduce this figure in a number of different ways, including as a stacked bar chart (see below). Using a stacked bar chart made it much more difficult to read the value of the different error bars, as well as making it almost impossible to see the contribution of excess change, due to it being overshadowed by the larger terms.

Instead, we have made changes to improve the readability of the original figure: we have removed the error bars and instead used shading to represent the standard deviation. We have also increased the size of the individual markers, so they are easier to see despite overlapping at some points. We have also changed the lines to a dotted line, to ensure the emphasis is on the individual data points and so is not suggesting a linear trend.

[Figure]

Line 255. We have noticed that the ship transects occurred at different seasons (see table A1). Do the authors think that the different temperatures, primary production, etc, of the different seasons could impact the results? In the abstract, the authors mentioned 'This remineralization-driven change may be caused by a change in the supply of biological material to depth…', We assume the large organic particles would have a different export rate in different seasons since it's a function of primary productivity and mixed layer depth. We would expect the authors to discuss a bit more about how they treat such variability.

While we expect the largest seasonal changes to occur in the upper 150 m of the water column, which we have excluded from our analysis, we believe any deeper seasonality is not large enough to alter our conclusions. The 1992 cruise was the only cruise to take place during summer, but since it is taken as a baseline to compute changes from year to year, while the different time of year may influence the quantitative differences between the years, it will not affect the year-to-year trends since they are all calculated from the same baseline. The 2004 cruise also occurred at a slightly different time of year, from April-May, that cruise sees the same trends as other years, with deoxygenation during 2004 due to excess and remineralisation changes being larger than in 1998, but smaller than those in 2010. We have now discussed seasonality within the discussion of the manuscript:

"The six cruises on the A05 transect generally took place between December and March, however two cruises did not occur during this time frame (1992 and 2004, see Table A1), meaning the results could be impacted by seasonal changes. To account for this, we excluded the upper 150 m from our analysis - the depths where we expect the greatest seasonality to occur and where they are clear in the initial temperature, DIC and oxygen profiles (Fig. 1b-d). The 1992 cruise occurred during summer, but since it was used as a baseline to calculate the changes between years, while this may lead to biases in the quantitative results, it should not affect the observed long-term trends. In the mechanisms that we see the clearest trends, i.e. in excess and remineralisation changes, the same trends were observed in 2004, suggesting that any seasonal changes deeper than 150 m are not large enough to impact the trends we see in the results." (Lines 320-327)

Line 270. The export ratio determines what fraction of productivity is exported, so a change in productivity does not necessarily have to lead to an equal change in export amount. It would be good to acknowledge this here and the consequences for the authors' conclusions.

Thank you for the suggestion. The following has been included in the discussion: "… there has been little change in primary production in the surface of the subtropical North Atlantic over the same time period (Macovei et al., 2019). This suggests that either there is has been change in the export ratio, with less of the productivity exported to depth, or that changes in large-scale circulation are driving the increased remineralisation." (Lines 293-295)

Line 285. 'The excess oxygen change has already doubled in magnitude between 1992 and 2015.' Can you report the value of this magnitude?

This has now been changed to:

"The excess oxygen change has already more than doubled in magnitude between 1998 and 2015, from -0.16 µmol kg$^{-1}$ to -0.47 µmol kg$^{-1}$ in the upper 150-500 m." (Lines 315-316)

Line 305. The conclusion part is too simple to summarize the key results of this study, including the methods used and the trend (magnitude, proportion, caveats, etc.) found from these GOSHIP transects. This is an important section; thus suggest the authors add more detail to this part.

The conclusions have now been extended to include further information about the method, and the observational data the method was used on. We also further discuss the magnitude of the remineralisation change and that we can only speculate and what is driving that change in remineralisation without further work.

Line 310: Where is Appendix A?

Appendix A contains only the table of information about the individual cruises. A section title has now been added to make this clearer.

---

## Author Response (AR2)

Thank you for the comments, we have now made the following changes, listed below in blue.

Line 70. Change "To cover the A05 section, we use a regular grid with" to " We then map the A05 sections onto a regular grid with ..

This has now been changed.

Caption Figure 1. Add c) to "The spatially-average b) potential .... c) DIC

This has now been corrected.

Inverse Method.

I suggest you incorporate section 2.3 into section 2.7, removing any repetition. By incorporating section 2.3 into section 2.7, it becomes clear that W1 and W2 are not randomly chosen, but they are the estimated parameter variance and variability of total parameter changes of the section, respectively.
Section 2.3 has now been moved into 2.7, with the final system of equations now described in Section 2.6.

Also, throughout the manuscript including Table 1. I suggest you replace "initial guess" with "a priori estimate" or "initial estimate". "Guess" suggests that that you have no knowledge, which is not true. You have used knowledge currently available to provide an informed initial estimate (and weights). This suggestion is driven by many conversations I have been involved in regarding inverse techniques and their validity/value.

Thank you for the suggestion, all instances of initial guess have now been changed to initial estimate.

Line 184-187 Change "Once the coefficients are calculated for each individual grid point along the A05 section, they form matrix A and the system in Equation 4 can be solved via a weighted least squares fit approach. The magnitude of the oxygen change terms at each point can then be added back into the equations relating oxygen change to temperature and DIC changes to compute the magnitude of each driver of temperature and DIC change at each point along the A05 section"
To "Once the coefficients are calculated for each individual grid point along the A05 section, they can be input into equations ##-## (add number of equation) and represent as system of simultaneous equations in the from W1Ax=W2b and solved via a weighted least squares fit approach. The magnitude of the oxygen change terms at each point can then be added back into the equations relating oxygen change to temperature and DIC changes to compute the magnitude of each driver of temperature and DIC change at each point along the A05 section."

This has now been changed, and is now at the start of subsection 2.6 due to merging subsection 2.3 and 2.7

Line 197-198. Change "In instances where no previous information is available, we instead assign an initial guess of zero, but must still give a non-zero value for the weighting in order to obtain a non-zero solution. In this case, when temporal variability data is not available, we assume an estimate of the spatial variability from previous literature to be comparable."
To
"In instances where no previous information is available, we assign an initial guess of zero and estimate the spatial variability from previous literature."
This has now been changed.

Figure 1. Are the contours that define the water masses potential density, neutral density or some other parameter? You need to include information on what the contours represent. The caption should be modified, for example assuming these contours are potential density; ".. with potential density (xx, xx, xx kg m-3) showing .."

The definitions of the water masses have now been added to the Fig. 1 caption:

"contours showing the upper (uNACW, $\sigma_0 < 26.7$ kg m$^{-3}$) and lower North Atlantic Central Water (lNACW, 26.7 kg m$^{-3} < \sigma_0 < 27.2$ kg m$^{-3}$), Antarctic Intermediate Water (AAIW, 27.2 kg m$^{-3} < \sigma_0 < 27.6$ kg m$^{-3}$), and upper North Atlantic Deep Water (uNADW, $\sigma_0 > 27.6$ kg m$^{-3}$ and $\sigma_2 < 37$ kg m$^{-3}$) based on definitions in Guallart et al. (2015) and labelled in g)."

Figure 6 Change "… from a) to rule out any effects due to seasonal variability." To "… from a) exclude effects due to seasonal variability."

This has now been changed.